# Feeding Black Pepper (*Piper nigrum*) or Exogenous Xylanase Improves the Blood Lipid Profile of Broiler Chickens Fed Wheat-Based Diets

**DOI:** 10.3390/vetsci10090587

**Published:** 2023-09-21

**Authors:** Vasil Radoslavov Pirgozliev, Stephen Charles Mansbridge, Isobel Margaret Whiting, Kristina Kljak, Artur Jozwik, Judith Maria Rollinger, Atanas Georgiev Atanasov, Stephen Paul Rose

**Affiliations:** 1National Institute of Poultry Husbandry, Harper Adams University, Newport TF10 8NB, UK; smansbridge@harper-adams.ac.uk (S.C.M.); iwhiting@harperadams.ac.uk (I.M.W.); sprose@harper-adams.ac.uk (S.P.R.); 2Department of Animal Nutrition, Faculty of Agriculture, University of Zagreb, 10000 Zagreb, Croatia; kkljak@agr.hr; 3Institute of Genetics and Animal Biotechnology, Polish Academy of Sciences, Jastrzebiec, 05-552 Magdalenka, Poland; aa.jozwik@igbzpan.pl (A.J.); tanas.atanasov@dhps.lbg.ac.at (A.G.A.); 4Department of Pharmaceutical Sciences, Division of Pharmacognosy, University of Vienna, A-1090 Vienna, Austria; judith.rollinger@univie.ac.at; 5Ludwig Boltzmann Institute for Digital Health and Patient Safety, Medical University of Vienna, A-1090 Vienna, Austria

**Keywords:** black pepper, xylanase, broilers, blood lipid profile, ME, digestibility, performance

## Abstract

**Simple Summary:**

Epidemiological research indicates that low blood plasma levels of high-density lipoprotein (HDL) and increased levels of low-density lipoprotein (LDL) are associated with an increased risk of cardiovascular diseases in humans. Yet, the impact of blood lipids on the health and wellbeing of poultry is not fully understood. This research provides information that feeding black peppercorn (BP) increases HDL and feeding xylanase (XYL) reduces LDL in the blood plasma of broiler chickens. For XYL-fed birds, this coincided with increased hepatic antioxidant capacity. Although the life span of commercial poultry is much shorter than that of humans, further understanding of the need to manipulate blood lipids for the improvement of the health and wellbeing of poultry is important. This study offers a nutritional approach to influence the production performance and health of poultry.

**Abstract:**

This study aimed to determine the impact of dietary black peppercorn (BP) and xylanase (XYL) alone or in combination on growth performance, dietary energy, nutrient digestibility and blood lipid profile when fed to male Ross 308 broiler chickens from the ages of 7 to 21 d. A wheat-soy-based basal feed that was formulated to be 0.42 MJ lower in metabolizable energy (ME) was mixed. The basal feed was then split into four batches, with the first batch set aside as the basal control; the second batch was supplemented with freshly milled BP; the third batch was supplemented with XYL; the fourth batch was supplemented with both BP and XYL, as in the previous two batches. Each diet was fed to eight pens, with two birds in a pen, following randomization. Feeding BP reduced bird growth and most of the digestibility coefficients but increased blood high-density lipoprotein (*p* < 0.05). Dietary XYL increased bird growth, dietary ME and nutrient digestibility (*p* < 0.05). In addition, XYL increased hepatic carotenoids and coenzyme Q_10_, but reduced blood low-density lipoprotein (*p* < 0.05). There were no BP by XYL interactions (*p* > 0.05) observed. Further research is needed to identify the optimum level of BP in broiler diets.

## 1. Introduction

The inclusion of dietary plant extracts to poultry diets is gaining popularity, not only for being used as an alternative to antimicrobial growth promotors [1] but also for their antioxidant properties [2] and potential health benefits [3]. Animal studies showed that the long-term intake of synthetic antioxidants, e.g., butylated hydroxyanisole and butylated hydroxytoluene [4,5], may lead to health risks including skin allergies, gastrointestinal tract (GIT) disorders and the increased likelihood of cancer [6,7]. Natural antioxidants, such as vitamin E and carotenoids, play important roles in maintaining poultry health and the productive and reproductive performance of breeders, layers, rearing birds and growing broilers [8]. Thus, studying the impact of antioxidants from natural origin, e.g., plant extracts from natural spices, on the health and welfare of poultry is an important aspect of nutritional research.

Black pepper (*Piper nigrum* L.) is a tropical plant belonging to the Piperaceae family and cultivated for its peppercorns (BP). It is native to parts of the Malabar Coast of India and is one of the earliest known spices. Black pepper is also well known for its use as a food ingredient and in traditional Chinese and Indian medicine [9]. The principal bioactive component of BP fruits, piperine, is an alkaloid that has a wide range of pharmacological effects, including antioxidant, anti-bacterial, anti-proliferative and anti-tumour and cholesterol-lowering properties [10]. Studies with rats have indicated that dietary BP can elevate the blood plasma levels of high-density lipoprotein (HDL) and decrease the plasma levels of total cholesterol (TC), triglyceride (TG) and low-density lipoprotein (LDL) cholesterol [11]. However, others [12] did not find changes in blood lipid variables attributed to feeding BP to broiler chickens. Previous research [9] showed that feeding BP to growing pigs increased HDL and vitamin C in blood. It has been reported [13] that feeding BP to poultry decreases blood TC. Dietary BP does not consistently affect the growth of broiler chickens, since [14,15] did not find enhanced growth performance, whilst dietary BP reduced the growth performance in other studies [16] or was found to be dose-dependent [13,17].

Exogenous xylanase (XYL) has been routinely used in poultry diets for over three decades. The mode of XYL action is mainly associated with a reduction in digesta viscosity, improved digestion and absorption of nutrients, dietary energy availability and the generation of fibres with prebiotic properties, leading to the improved growth performance of broilers [18,19]. Beyond performance results, it has been reported [20] that dietary XYL increases blood HDL and reduces TC, although it was not identified previously [21]. Recently, it has been reported that dietary XYL increases hepatic vitamin E [22] and coenzyme Q_10_ [23] when fed to broiler chickens, suggesting antioxidant properties. In addition, there is lack of information on the interaction between dietary BP and supplementary XYL.

Blood mineral composition can be affected by diet, bird strain, sex, age and season [24], although information on birds fed BP and XYL is limited. No difference was found in blood sodium and potassium concentrations of birds fed BP [12]. An increase in blood phosphorus in broilers fed a mixture of phytase and XYL was found, but no differences in calcium level was found [25]. Whilst this limited information is available, there is a general lack of knowledge on the interaction between BP and XYL when fed to broiler chickens. Since XYL is known to modulate nutrient digestibility and absorption, including minerals, in combination with the bioactive properties of BP, an interaction may be expected. Thus, the experiment aimed to investigate the effect of dietary BP and XYL, alone or in combination, on growth performance, relative GIT organ weight, the availability of dietary energy and nutrients, hepatic antioxidant concentration, blood lipids and mineral profiles of broiler chickens. To avoid confounding with antioxidants in maize, a wheat-based diet was chosen [20,21].

## 2. Materials and Methods

### 2.1. Experimental Diets and BP Sample

A wheat-soy-based basal grower feed that was approximately 0.42 MJ (100 kcal) lower than the breeder’s recommendations (Aviagen Ltd., Edinburgh, UK) was mixed for the experiment (Table 1). Black peppercorns (registered feed material: ID number 009237-EN) with a determined amount of 42 g/kg piperine were purchased from the market (Sainsburys, UK) (Table 2).

The basal feed was then split into four batches, with the first batch set aside as the basal control portion; the second batch was supplemented with an Aspergillus oryzae commercial preparation of endo-1,4-beta-XYL at 200 FXU/kg (Ronozyme WX (CT), DSM, Kaiseraugst, Switzerland); the third batch was supplemented with a 10 g/kg diet of freshly milled BP corns; the fourth batch was supplemented with both XYL and BP, as in the previous two batches. All diets contained 20 g/kg of acid-insoluble ash (AIA), a feed-grade diatomaceous earth (Multi-Mite^®^, Wiltshire, UK), as an insoluble marker. Feed was not reformulated to account for XYL or BP inclusion.

### 2.2. Birds, Management and Sample Collection

Eighty male Ross 308 chicks were purchased at a day old from a commercial hatchery (Cyril Bason Ltd., Craven Arms, UK). On arrival, all chicks were allocated to a single-floor pen, reared on wood shavings and fed a proprietary broiler feed. On day 7, sixty-four birds were allocated to 32 pens, two birds per pen, following randomization. Each pen (experimental unit) was equipped with a trough feeder and drinker. The room temperature was 32 °C when the chicks were a day old and was gradually reduced to 20 °C by the end of the study (21 d old). A standard lighting programme for broilers was used (Aviagen Ltd., Edinburgh, UK). For the last three days of the study, from the ages of 18 to 21 d, the solid floor in each pen was replaced with wire mesh, and excreta were collected each day. Pooled excreta were oven-dried at 60 °C and milled prior to chemical analysis. Data on growth performance, including feed intake (FI), weight gain (WG) and mortality-corrected feed conversion ratio (FCR), were obtained from 7 d to 21 d of age. At the end of the study, at 21 d of age, one bird per pen was head-only electrically stunned, and blood was collected during exsanguination (terminal blood collection) into lithium heparin tubes. Plasma was obtained via centrifugation of the blood and stored at −80 °C until analysis. The liver of the dead bird was weighed and immediately stored at −20 °C until analysis. The organs from the GIT of the same bird, including proventriculus and gizzard (PG), duodenum (D), pancreas (P), jejunum (J), ileum (I), caeca and the spleen, were weighed and processed as previously described [26].

### 2.3. Laboratory Analyses

Piperine in BP was determined at the Department of Pharmacognosy, University of Vienna (Austria) [27]. Starch and non-starch polysaccharides (NSPs) in basal feed and BP were determined by Englyst Carbohydrates Ltd. (Southampton, UK), following standard methodology [28,29]. Dry matter (DM), crude protein (CP; 6.25 × N) and oil (crude fat, CF) in feed and excreta samples were determined following standard procedures as previously described [30]. The gross energy (GE) in feed, BP and excreta samples was determined using a Parr 6200 isoperibol oxygen bomb calorimeter (Parr Instrument Co., Moline, IL, USA). Neutral detergent fibres (NDFs) in samples were determined as previously described [31]. Acid-insoluble ash (AIA) in feed and excreta was determined by following the standard procedure [32]. The antioxidants in basal feed, BP and liver, including vitamin E, coenzyme Q_10_ and total carotenoids (TCS), were determined as previously described [33,34]. Blood plasma analyses including total cholesterol (TC), high-density lipoprotein (HDL), low-density lipoproteins (LDL), triglyceride (TG), total protein (TP), Ca, P, Fe and Mg were also performed [9,35,36].

### 2.4. Statistical Analysis

Data analyses were performed in Genstat (23rd edition) statistical software (IACR Rothamstead, Hertfordshire, UK). Comparisons among the studied variables were performed via two-way ANOVA using a 2 × 2 factorial design (dietary BP × xylanase). Outliers, homogeneity and normality of residuals were assessed prior to the ANOVA. In all instances, differences were reported as significant at *p* < 0.05.

## 3. Results

The determined chemical composition of the basal diet and BP are presented in Table 2. Black pepper contained less CP, but more CF, GE, starch, carotenoids and coenzyme Q_10_ compared to the basal diet. The total NSP content in the BP and the basal diet was similar (Table 2).

The results of bird growth performance, AMEn and total tract nutrient digestibility coefficients are presented in Table 3.

Birds fed the BP diet had lower final BWs (*p* < 0.001), lower daily FIs (*p* < 0.05), lower WGs (*p* < 0.001), higher FCRs (*p* < 0.001) and lower DMD, ND and NDFD values (*p* < 0.05) compared to those fed BP-free diets. Dietary BP did not influence dietary AMEn and FD (*p* > 0.05). Feeding XYL improved the final BW, daily FI, daily WG, AMEn, DMD, ND and NDFD coefficients (*p* < 0.05). There was no diet by XYL interaction observed (Table 3).

Table 4 presents information on relative GIT organ weights. Feeding BP increased (*p* < 0.05) the relative weight of D and J.

Whilst dietary XYL overall did not significantly affect the weight of SI, when fed alone (without BP), it reduced the relative weight of SI (*p* < 0.05). Similarly, feeding dietary XYL alone reduced the relative weight of GIT compared to the combination of BP and XYL (*p* < 0.05); however, this was not different to the basal control or BP-only diets.

Dietary XYL increased the hepatic concentration of TCS and coenzyme Q_10_ (*p* < 0.05) (Table 5).

There was a tendency of BP by XYL interaction (*p* = 0.067) as feeding diet containing both, BP and XYL, numerically increased hepatic vitamin E concentration (basal feed, 74 µg/g; BP feed, 75 µg/g; XYL feed, 63 µg/g; BP and XYL feed, 88 µg/g; SEM = 6.34).

Feeding BP resulted in increased HDL (*p* < 0.05) and reduced Ca (*p* < 0.05) in blood plasma (Table 6). Feeding XYL reduced (*p* < 0.05) blood LDL level. No differences were found for other studied variables in blood (*p* > 0.05). There was no BP by XYL interaction observed.

## 4. Discussion

The chemical composition of BP was within the expected range [9,27,37]. It is recognized that the composition of plants, e.g., wheat [38], naturally varies due to different climate, soil, agronomy, cultivars, geographical regions, processing and laboratory analysis techniques. The popularity of foods with high antioxidant content is increasing; thus, the observed levels of coenzyme Q_10_ (103 µg/g) in BP may be used to enhance animal and human diets. Several different diseases have been associated with pro-oxidative processes [39,40], and the improvement of antioxidant defences of the body has been considered as a preventive and therapeutic strategy [41,42].

Chickens fed non-BP-containing diets had 12% lower body weight than breeders’ recommendations, which was expected and can be explained by them being fed mash rather than pelleted feed and kept in small groups [43,44], although this was not considered to be detrimental to the experimental aims. However, birds fed BP had an average body weight that was 31% lower compared to breeders’ recommendations, which is in addition to a 21.4% reduction in FI compared to non-BP-fed birds. Previous research also showed a sort of dose-dependent manner in the response of FI and growth to dietary BP [17]. In general, lower BP inclusion, i.e., about 5 g/kg, did not have an impact, although the inclusion of 10 g/kg or more led to reduced FI and overall growth performance [13,16]. Birds’ feed intake is affected primarily by feed’s visual and textural properties, but taste and smell can also influence it [45,46]. Piperine, the main active ingredient in BP, is a naturally occurring pungent and spicy constituent of BP and may bring more intensive dietary flavour when added in greater levels [47]. A study feeding piperine to rats [48] reported reduced FI and WG in those fed 50 mg/kg body weight. Additionally, it has been reported that the introduction of as little as 20 mg of piperine per kg in the diet inhibited the volume of gastric juice, gastric acidity, and pepsin of rats and mice [49]. In the present study, the birds fed BP were consuming 58.2 g of feed daily (14-day feeding period), containing 0.58 g of BP or about 24 mg of piperine, which is about 46 mg/kg of the average daily body weight (523 g). Thus, indicating that the dietary level of piperine in the reported study was high for rats and may therefore also be potentially high for chickens, though further research is required to confirm the tolerances of chickens to piperine. These high levels of piperine may be the reason for the observed lower digestibility and performance results.

In agreement with recent studies [19,23,50], dietary XYL increased the final BW, daily FI, daily WG, AMEn and nutrient digestibility coefficients. Exogenous XYL has been routinely used in poultry nutrition to hydrolyze NSP and improve the feeding quality of fibre-rich diets [18,51]. The beneficial effect of XYL in wheat-based diets is mainly attributed to the reduction in digesta viscosity, improving digestion and absorption of nutrients, dietary energy availability and the subsequent growth performance of chickens [52,53]. The observed differences in the enhancement of bird growth performance, available energy and digestion in the literature are attributed to variations in dietary fibre [38], birds’ age and study duration [44], the transit rate of digesta in various segments of the broilers’ GIT [54] and rearing environment [55].

Although feed materials are the main determinant of antioxidant composition in the liver [34], feed supplements/additives other than antioxidants (e.g., phytase and XYL) can affect the efficiency of antioxidant assimilation from the diet and subsequently their accumulation in the liver [3,56,57]. An increase in hepatic vitamin E and coenzyme Q_10_ content, respectively, when feeding XYL to broilers has been also reported [22,23]. The mode of action of XYL accounting for the observed increase in hepatic carotenoids and coenzyme Q_10_ in XYL fed birds is unclear. A high concentration of coenzyme Q_10_ in cell membranes enhances their antioxidant status against lipid peroxidation. In agreement with previous research [23], the observed increase in hepatic antioxidants suggests that dietary XYL improves not only dietary energy and nutrient availability, but also the antioxidant status within commercial poultry. The increased AMEn, DMD, ND and NDFD of the birds fed XYL diets in the present study is coupled with reduced SI weights. In general, if the efficiency of digestion is consistently suboptimal, whether due to ingredient quality, microbial interaction or anti-nutritive factors, the GIT responds by increasing in both size (surface area) and digestive enzyme output [58].

The results of blood plasma lipid fractions, TP, Ca and P were within the expected range for broiler chickens [24]. The responses in HDL and LDL in blood are supported by previous research with BP in rats [11], pigs [9] and with XYL in broilers [20]. Epidemiological research shows that low blood levels of HDL and high levels of LDL are associated with an increased risk of cardiovascular diseases in humans [59,60]. There are similarities in lipid metabolism between chickens and humans that make birds a suitable model for investigation [61]; thus, a balance between blood lipids may also be important for their health and wellbeing. Sudden death syndrome (SDS), one of the main reasons for mortality in growing broilers, is also associated with heart problems (acute heart failure) as a potential cause of the disease in broiler chickens [62]. However, the use of BP and XYL in the reported study elicited favourable responses in levels of HDL and LDL; thus, they may have the potential to decrease the incidence of SDS in broilers. It can be also speculated that the inclusion of BP and XYL to broiler diets, alone or in combination, may not only enhance the health of the birds but may also reduce the unhealthy lipid fraction in meat, bringing further benefits to consumer health. Low-density lipids and HDL are two main classes of lipoproteins synthesized and secreted by the liver. The triglycerides are primarily used for LDL and most of the phospholipids and cholesterol are in HDL [63]. It has been found that changes in LDL and HDL production may relate to fatty liver haemorrhagic syndrome in laying hens [64,65]. Thus, further research on feeding BP and other plant extracts on bird health is warranted.

## 5. Conclusions

The inclusion of 10 g/kg of BP to broiler diets increased blood plasma HDL but reduced production performance, including the growth rate of the birds. Dietary XYL, however, reduced blood plasma LDL and increased hepatic antioxidants concentrations, as well as improved bird growth and the availability of dietary energy and nutrients. The lack of interaction between BP and XYL shows that when fed together, these supplements do not cause detrimental effects. Further work is needed to establish the optimum level of dietary BP for poultry; however, XYL can be fed as standard.

## Figures and Tables

**Table 1 vetsci-10-00587-t001:** Composition of broiler chicken basal feed (g/kg, as-fed) used in the experiment.

Ingredients (g/kg)	Basal Feed
Wheat	651.0
Soybean meal (48% CP)	219.7
Soybean meal (full fat)	50.0
Vegetable oil	20.0
Dicalcium phosphate	14.5
Limestone	12.5
NaCl	1.7
Lysine	2.7
Methionine	3.9
Vitamin/mineral premix ^1^	4.0
Acid-insoluble ash	20.0
	100
Calculated analysis (as fed):	
Crude protein g/kg	206
ME MJ/kg	12.67
Crude fat g/kg	44.4
Ca g/kg	9.7
Available P g/kg	4.6
Lysine g/kg	12.4
Methionine + cysteine g/kg	9.9

^1^ The vitamin and mineral premix contained vitamins and trace elements to meet the breeder’s recommendation (Aviagen Ltd., Edinburgh, UK). The premix provided is as follows (units/kg diet): retinol 3600 μg, cholecalciferol 125 μg, α-tocopherol 34 mg, menadione 3 mg, thiamine 2 mg, riboflavin 7 mg, pyridoxine 5 mg, cobalamin 15 μg, nicotinic acid 50 mg, pantothenic acid 15 mg, folic acid 1 mg, biotin 200 μg, iron 80 mg, copper 10 mg, manganese 100 mg, cobalt 0.5 mg, zinc 80 mg, iodine 1 mg, selenium 0.2 mg and molybdenum 0.5 mg.

**Table 2 vetsci-10-00587-t002:** Determined chemical composition of basal feed and black pepper (as-fed basis).

Determined Values	Basal Feed	Black Pepper
Dry matter (g/kg)	902	921
Crude protein (g/kg)	218	130
Crude fat (g/kg)	35	55
Gross energy (MJ/kg)	15.51	17.15
Piperine (g/kg)	Nd	42
Starch (g/kg)	429	491
Soluble non-starch polysaccharides (NSPs, g/kg)	36	9
Insoluble non-starch polysaccharides (NSPins, g/kg)	60	83
Total non-starch polysaccharides (NSPt, g/kg)	96	92
Neutral detergent fibres (NDF, g/kg)	84	Nd
Total carotenoids (µg/g)	0.066	2.777
Vitamin E (µg/g)	21.2	Nd
Coenzyme Q_10_ (µg/g)	2	103

MJ = megajoule; Nd = not determined.

**Table 3 vetsci-10-00587-t003:** Effect of dietary black pepper (BP) and xylanase (XYL) on start and end body weight (BW), daily feed intake (FI), weight gain (WG), mortality-corrected feed conversion ratio (FCR), N-corrected apparent metabolizable energy (AMEn), total tract dry matter (DMD), nitrogen (ND), fat (FD) and neutral detergent fibre (NDFD) digestibility coefficients when fed to broiler chickens from 7 to 21 d of age ^1^.

Treatment	BW 7 d (g)	BW 21 d (g)	FI (g/b/d)	WG (g/b/d)	FCR (g:g)	AMEn (MJ/kg DM)	DMD	ND	FD	NDFD
BP										
No	175	897	65.7	51.6	1.276	12.95	0.734	0.652	0.653	0.247
Yes	179	702	58.2	37.4	1.567	12.76	0.716	0.612	0.666	0.190
XYL										
No	175	767	59.3	42.2	1.438	12.71	0.719	0.616	0.645	0.190
Yes	178	833	64.6	46.8	1.404	13.00	0.732	0.647	0.674	0.247
SEM	1.8	16.6	1.46	1.17	0.0259	0.092	0.0045	0.0074	0.0233	0.0177
Probabilities										
BP	0.115	<0.001	0.006	<0.001	<0.001	0.167	0.012	0.008	0.701	0.033
XYL	0.298	0.010	0.019	0.012	0.365	0.039	0.048	0.041	0.387	0.036
BP × XYL	0.449	0.396	0.667	0.438	0.490	0.311	0.192	0.792	0.394	0.889

^1^ Each mean represents values from sixteen replicate pens for main effects; AMEn and digestibility coefficients were determined between 18 and 21 d of age; SEM = pooled standard errors of mean.

**Table 4 vetsci-10-00587-t004:** Effect of dietary black pepper (BP) and xylanase (XYL) on the relative organ weight expressed as the percent of bird body weight (BW) of gastrointestinal tract (GIT), including proventriculus and gizzard (PG), duodenum (D), jejunum (J), ileum (I), small intestine (SI), caeca, pancreas, liver and spleen of 21 d old broiler chickens ^1^.

Treatment	BW (g)	PG	D	J	I	SI	Caeca	Pancreas	GIT	Liver
BP										
No	911	2.97	1.64	2.61	2.14	6.39	0.75	0.44	10.54	3.57
Yes	692	2.78	1.86	2.86	2.17	6.89	0.72	0.43	10.81	3.82
XYL										
No	780	2.84	1.69	2.79	2.21	6.69	0.72	0.44	10.68	3.73
Yes	823	2.91	1.81	2.68	2.11	6.59	0.75	0.43	10.68	3.66
SEM	-	0.110	0.051	0.066	0.069	0.133	0.027	0.021	0.165	0.149
BP XYL										
No No	882	3.03	1.61	2.75	2.31 ^a^	6.67 ^a^	0.72	0.47	10.88 ^ab^	3.74
No Yes	940	2.91	1.68	2.47	1.97 ^b^	6.11 ^b^	0.77	0.41	10.20 ^a^	3.39
Yes No	679	2.64	1.78	2.83	2.10 ^ab^	6.71 ^a^	0.71	0.41	10.47 ^ab^	3.71
Yes Yes	706	2.91	1.93	2.90	2.24 ^ab^	7.07 ^a^	0.72	0.45	11.15 ^b^	3.92
	-	0.155	0.072	0.066	0.098	0.189	0.039	0.030	0.234	0.210
Probabilities										
BP	-	0.357	0.008	0.014	0.741	0.016	0.431	0.690	0.269	0.246
XYL	-	0.838	0.138	0.267	0.311	0.611	0.402	0.750	0.997	0.738
BP × XYL	-	0.341	0.596	0.071	0.023	0.024	0.617	0.149	0.009	0.204

^1^ Each mean represents values from sixteen replicate pens for main effects; GIT = gastrointestinal tract weight without liver; SEM = pooled standard errors of mean; means with different superscripts are statistically significant (*p* < 0.05). ^a,b^ Statistically significant (*p* < 0.05) difference.

**Table 5 vetsci-10-00587-t005:** Effect of dietary black pepper (BP) and xylanase (XYL) on hepatic carotenoids, vitamin E and coenzyme Q_10_ of 21 d old broiler chickens ^1^.

Treatment	Carotenoids (µg/g)	Vitamin E (µg/g)	Coenzyme Q_10_ (µg/g)
BP			
No	1.44	69	418
Yes	1.49	81	402
XYL			
No	1.33	75	375
Yes	1.60	76	444
SEM	0.073	4.5	20.8
Probabilities			
BP	0.668	0.060	0.603
XYL	0.015	0.836	0.030
BP × XYL	0.525	0.067	0.646

^1^ Each mean represents values from sixteen replicate pens for main effects; SEM = pooled standard errors of mean.

**Table 6 vetsci-10-00587-t006:** Effect of dietary black pepper (BP) and xylanase (XYL) on blood plasma variables of 21 d old broiler chickens ^1^.

Treatment	TC (mmol/L)	HDL (mmol/L)	LDL (mmol/L)	TRIGL (mmol/L)	TP (g/L)	Ca (mmol/L)	P (mmol/L)
BP							
No	4.1	2.9	0.55	1.39	22.6	3.3	2.10
Yes	4.4	3.1	0.61	1.37	24.0	3.0	1.88
XYL							
No	4.3	3.0	0.66	1.37	23.3	3.2	2.14
Yes	4.1	3.0	0.51	1.39	23.3	3.1	1.84
SEM	0.133	0.07	0.037	0.141	0.67	0.07	0.1145
Probabilities							
BP	0.177	0.017	0.278	0.916	0.174	0.022	0.176
XYL	0.401	0.918	0.010	0.916	0.953	0.947	0.073
BP × XYL	0.500	0.857	0.103	0.566	0.343	0.979	0.429

^1^ Each mean represents values from sixteen replicate pens for main effects; TC = total cholesterol; HDL = high-density lipoproteins; LDL = low-density lipoproteins; TG = triglyceride; TP = total protein; Ca = calcium; P = phosphorus; SEM = pooled standard errors of mean.

## Data Availability

The data that support the findings of this study are available on reasonable request from the corresponding author.

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
