# Peer review of "Feeding Black Pepper (Piper nigrum) or Exogenous Xylanase Improves the Blood Lipid Profile of Broiler Chickens Fed Wheat-Based Diets"

_vetsci, 2023, doi:10.3390/vetsci10090587_

Round 1

Reviewer 1 Report

1. Insufficient animal sample size for feeding trail (totally 64 broilers).

2. The title did not focus on the topic. In fact, the additive used in this project is under the condition of feeding broilers with wheat-based diet, this is important.

3. Also, the Introduction should reorganization, why use these two additives when broilers fed wheat-based diet?

4. M&M: I can't clearly find the added dose, experimental design Too much redundant information.

5. Discussion: the same problem as comment 3.

Author Response

  1. Insufficient animal sample size for feeding trail (totally 64 broilers).

Response by V Pirgozliev – The floor of the pens was approximately 0.12-0.14 square metres and did not permit more birds to be involved in the study. The experimental unit was a pen with 2 birds, thus providing 8 experimental units per diet fed. Although we appreciate that more birds may reduce variation, the number of replications is more important, and we believe that the relatively small number of birds was not detrimental to the experimental aims. We have previously performed successful experiments with this facility using the same experimental design.

  1. The title did not focus on the topic. In fact, the additive used in this project is under the condition of feeding broilers with wheat-based diet, this is important.

Response by V Pirgozliev – Title now changed “Feeding Black Pepper (Piper nigrum) or Exogenous Xylanase Improves the Blood Lipid Profile of Broiler Chickens fed wheat-based diets”.

  1. Also, the Introduction should reorganization, why use these two additives when broilers fed wheat-based diet?

Response by V Pirgozliev – Than you for the suggestion. Now explained in Introduction wheat-based diets was used to avoid confounding with antioxidants in maize.

  1. M&M: I can't clearly find the added dose, experimental design Too much redundant information.

Response by V Pirgozliev – Dietary black pepper (at 10 g/kg) and exogenous xylanase (at 200 FXU units/kg) were added on the top of the basal control and those diets were not reformulated to account for XYL or BP inclusion. The amount was so small and did not change basal diet formulation. All explained in M&M section. To our experience, MDPI journals count as plagiarism any sentence repetition, including methods of analysis. To avoid it in this paper we prefer to describe what had been done in this way. The reader still can find the information needed.

  1. Discussion: the same problem as comment 3.

Response by V Pirgozliev – Information on the impact of XYL in wheat-based diets available in Discussion section.

Reviewer 2 Report

The paper titled "Feeding Black Pepper (Piper nigrum) or Exogenous Xylanase Improves the Blood Lipid Profile of Broiler Chickens" sheds light on an intriguing avenue for improving the health and lipid metabolism of broiler chickens. The introduction of the paper effectively sets the stage by highlighting the growing concerns surrounding lipid metabolism in broiler chickens and the potential consequences for both the birds' well-being and the quality of their products for human consumption.  

One of the paper's strong points is the thorough description of the experimental design and methodology. The inclusion of essential details, such as the composition of the diets, the duration of the study, and the parameters measured, contributes to the paper's credibility and reproducibility. The results section presents a well-structured account of the findings, effectively using tables to display the data. The reported improvements in blood lipid profiles due to both black pepper supplementation and exogenous xylanase administration are promising. However, some aspects could be further strengthened to enhance the impact of the paper. The authors need to explain the novelty of the study. Many studies have been conducted on the topic, what is the novelty of the topic they have chosen to explore need to be the point of acception of this paper.

Conclusion is needed to ba added in the abstract. 

I see that English edit is must for the paper. 

In materials and methods the level of Nacl in the diet table is too high please check. further it is not known whether the Table 1 provides the starter or finisher phase dietary plan. The abbreviations in Table 2 must be defined below the table. Why one level of black paper seeds was used. Why not multiple levels, How and why the authors were confident that this level would work? was adaptation period given. it is not clear

English of the paper should be improved in places. 

Author Response

The paper titled "Feeding Black Pepper (Piper nigrum) or Exogenous Xylanase Improves the Blood Lipid Profile of Broiler Chickens" sheds light on an intriguing avenue for improving the health and lipid metabolism of broiler chickens. The introduction of the paper effectively sets the stage by highlighting the growing concerns surrounding lipid metabolism in broiler chickens and the potential consequences for both the birds' well-being and the quality of their products for human consumption.  

One of the paper's strong points is the thorough description of the experimental design and methodology. The inclusion of essential details, such as the composition of the diets, the duration of the study, and the parameters measured, contributes to the paper's credibility and reproducibility. The results section presents a well-structured account of the findings, effectively using tables to display the data. The reported improvements in blood lipid profiles due to both black pepper supplementation and exogenous xylanase administration are promising. However, some aspects could be further strengthened to enhance the impact of the paper. The authors need to explain the novelty of the study. Many studies have been conducted on the topic, what is the novelty of the topic they have chosen to explore need to be the point of acception of this paper.

Response by V Pirgozliev – We believe that the novelty of the experiment is studying the interaction between dietary BP and supplementary XYL. Now stated in Introduction and Discussion.

Conclusion is needed to be added in the abstract. 

Response by V Pirgozliev – The problem is that we have word limit in Abstract and we are not able to write much there. I hope that the Simple Summary is complementing the info in Abstract.

I see that English edit is must for the paper. 

In materials and methods the level of NaCl in the diet table is too high please check.

Response by V Pirgozliev – Thanks for the comment. It was an error. Now corrected.

Further it is not known whether the Table 1 provides the starter or finisher phase dietary plan.

Response by V Pirgozliev – In this experiment we used the diet that is on Table 1. We used proprietary broiler starter feed in pre-study period and we do not display it because is not a part of this experiment.

The abbreviations in Table 2 must be defined below the table.

Response by V Pirgozliev – Now explained under Table 2 that MJ is for megajoule. I believe we no need to explain the abbreviations of g, kg and µg.

Why one level of black paper seeds was used. Why not multiple levels, How and why the authors were confident that this level would work? was adaptation period given. it is not clear.

Response by V Pirgozliev – The aim of this experiment was to determine the impact of dietary Black peppercorn and xylanase alone or in combination on growth performance, dietary energy, nutrient digestibility and blood lipids profile when fed to broiler chickens. We did not aim to compare the impact of graded level of black pepper on the studied variables. It has been done by others. We chose the level based on literature search. In research you are never confident if something will “work” and if it works why do it again?

Comments on the Quality of English Language

English of the paper should be improved in places. 

Response by V Pirgozliev – My co-authors Drs SC Mansbridge, IM Whiting and SP Rose are original English speakers, and they carefully check the paper. Please suggest which parts of the papers need to be improved regarding English language.

Reviewer 3 Report

The results section, especially the tables could be presented in a clearer manner. Instead of listing the BP (yes, no) and XYL (yes, no) list the four treatment and the contrasts. this would be less confusing. 

In conclusion, line 286-289- I don't think it is appropriate to say may bring additional benefits, it would be better to say, 'did not cause detrimental effects.'

Check grammar lines 66 and 67.

Check grammar lines 227 and 228.

Author Response

The results section, especially the tables could be presented in a clearer manner. Instead of listing the BP (yes, no) and XYL (yes, no) list the four treatment and the contrasts. This would be less confusing. 

Response by V Pirgozliev – If there is an interaction between dietary black pepper and xylanase we provided the values (see Table 4). If there is no interaction, as in the rest of the tables we provide the mean values only. This is a standard practice how to describe stats in research papers when you use factorial design.

In conclusion, line 286-289- I don't think it is appropriate to say may bring additional benefits, it would be better to say, 'did not cause detrimental effects.'

Response by V Pirgozliev – Changed in text.

Comments on the Quality of English Language

Check grammar lines 66 and 67.

Response by V Pirgozliev – Changed in text.

Check grammar lines 227 and 228.

Response by V Pirgozliev – Changed in text.

Reviewer 4 Report

The heading of Table 4 repeats that of Table 3, so I can't fully analyse the work. Therefore, I request that the authors resend the work with this correction.

Author Response

The heading of Table 4 repeats that of Table 3, so I can't fully analyse the work. Therefore, I request that the authors resend the work with this correction.

Response by V Pirgozliev – Thank you very much for the comment. Now changed in Table 4 headings.

Round 2

Reviewer 1 Report

none